# Predictors of community pharmacists' readiness to implement deprescribing of inappropriate medications for older adults in Qatar

Marwa Elshazly◉, Sondus Jawad◉, Ayesha Ahmed◉, Hager ElGeed,
Kazeem Babatunde Yusuff◉ *

Department of Clinical Pharmacy and Practice, College of Pharmacy, QU Health Sector, Qatar University, Doha, Qatar

◉ Authors contributed equally to the study.
* kyusuff@qu.edu.qa; yusuffkby@yahoo.co.uk

⬔ OPEN ACCESS

## Abstract

There is a paucity of studies focused on the predictors of community pharmacists' readiness to deprescribe inappropriate medications for older adults especially in developing settings. The study aimed to use the situational theory of leadership to determine community pharmacists' readiness to implement deprescribing of inappropriate medications for older adults, and as well as its significant predictors. A theory-driven cross-sectional assessment of the readiness (knowledge and confidence) of 252 community pharmacists was conducted in Qatar with a pre-tested 40-item questionnaire. Knowledge and confidence were assessed with a 2-point and 4-point Likert-type scale respectively. The maximum obtainable score for readiness was 16. Readiness was categorized as high (≥ median) or low (<median). Bivariate logistic regression was used to identify the significant predictors. The response rate was 79.4% (200). The majority of the community pharmacists were females (54.5%), within the age range of 20–40 years (88.0%), had BSc / BPharm as the highest educational qualification (70.5%), were full-time employee (97.0%) and consisted mainly of 5 nationalities (91.0%). The median (IQR) readiness score for community pharmacists was 13 (4) (minimum = 4, maximum = 16). Overall, 54.4% (109) of community pharmacists reported high confidence score (≥ median), while 60.5% (121) reported high knowledge score (≥ median). Readiness score was significantly higher among female community pharmacists (p = 0.048) and respondents who reported exposure to deprescribing in undergraduate training (p = 0.001). Overall, community pharmacists felt knowledgeable and confident to implement deprescribing of inappropriate medicines for older adults. However, a number of critical readiness gaps requiring educational interventions especially regarding how to access and use deprescribing toolkits and algorithms were identified. The most significant predictors of community pharmacists' readiness were gender (female) and exposure to deprescribing in undergraduate training.

**Data availability statement:** All relevant data are within the manuscript.

**Funding:** This work was supported by an Undergraduate Research Experience Program (UREP) award [UREP29-092-3-029] from the Qatar National Research Fund (a member of The Qatar Foundation). The contents are solely the responsibility of the authors.

**Competing interests:** The authors have declared that no competing interests exist.

## Introduction

The achievement of optimal outcomes that meet patient-specific therapeutic needs and reduce exposure to potentially harmful medication-related problems is predicated on rational use of medicines [1]. This is particularly important for older adults who often present with multiple chronic medical conditions requiring long term multiple drug therapy that put them at higher risk of polypharmacy, potentially inappropriate medications (PIMs), and the associated adverse outcomes [2–8]. For instance, an estimated 29% − 31% of elderly patients in the US and Canada were prescribed inappropriate medications that are considered potentially harmful and have safer alternatives [9–10]. This pales in comparison to the findings reported from two recent studies conducted in a Middle Eastern Gulf setting such as Qatar that put the prevalence of PIMs and polypharmacy among older adults at 60.7% and 75.5% respectively [11–12].

Deprescribing is a systematic evidence-based process that has gained ascendancy in contemporary times [13]. This is notwithstanding the lack of consensus regarding the succinct definition of deprescribing. However, Reeve et al, in a seminal paper proposed a definition of deprescribing as "*the process of withdrawal of an inappropriate medication, supervised by a health care professional with the goal of managing polypharmacy and improving outcomes*" [14]. Hence, deprescribing provides an opportunity for a thorough review of the appropriateness of medications prescribed for older adults, the identification of medications which are no longer required or harmful, and recommendation of outright discontinuation, gradual withdrawal or switching to safer and more effective options [15–16]. The deprescribing process involves the use of validated toolkits and algorithms that are evidence-based clinical guide detailing steps that must be followed in the assessment and recommendation of specific medications for deprescribing [17–20].

Community pharmacists are trusted healthcare professional with specialized expertise to optimize prescribing for older adults especially at the primary care level. They are particularly well-suited for these roles due to their geographic availability and ease of social access. This ease of access provides an excellent opportunity for older adults to access a functional medicine optimization service such as deprescribing that may enhance the achievement of optimal outcomes [21]. Several published studies done mostly in developed settings showed that community pharmacists possess the knowledge and confidence to implement deprescribing of inappropriate and potentially harmful medications for older adults [22–23]. Indeed, a systematic review of community pharmacists' role in deprescribing by Buzancic et al, showed that they can competently lead deprescribing interventions, collaborate effectively with physicians and patients to provide an effective deprescribing service that ensures optimal outcomes [24]. In addition, a recent study conducted in Ireland reported that community pharmacists are knowledgeable, confident and ready to identify and deprescribe the use of inappropriate medications for elderly patients [25].

However, studies focused on the assessment of community pharmacists' knowledge and confidence to deprescribe inappropriate medications especially for older adults in a developing Middle Eastern Gulf setting, including Qatar are scanty. Indeed,

literature search revealed only one such study conducted in the United Arab Emirates (UAE) which reported that only 25.3% of respondents were adjudged as having a good knowledge of medication classes that are good candidates for deprescribing [26]. However, the UAE study was not foregrounded by any theoretical framework that ensures a holistic assessment of readiness. The current study was foregrounded by a key component of the situational theory of leadership, followers' readiness or maturity, which comprises of two components including confidence and knowledge [27]. The choice of this theoretical framework is appropriate as it assesses the readiness of employees to successfully complete assigned tasks, and generate data that can be used to design training and development interventions focused on maximizing staff and organizational performance. The readiness assessment will result in the classification of employees into four categories including: high knowledge and high confidence; high knowledge and low confidence; low knowledge and high confidence; and low knowledge and low confidence [27]. This classification will enhance the precise identification of employee-specific interventions that are needed to improve readiness to complete assigned tasks. The use of this theoretical framework for the assessment of community pharmacists to implement deprescribing for older adults is being reported for the first time.

Furthermore, the provision of a deprescribing service guided by a structured framework is currently not within the scope of practice for community pharmacists generally in the Middle Eastern Gulf countries. Indeed, a recent study reported a number of perceived barriers to the implementation of community pharmacist-led deprescribing in Qatar [28]. Hence, a baseline assessment of the readiness of community pharmacists to implement a deprescribing service especially for a vulnerable group such as older adults is the appropriate starting point. Hence, the current study may provide new significant perspectives that will not only add to global knowledge in the research area but may also provide insights that can be used to develop appropriate interventions to fill any probably readiness gaps. The primary objective of the study was to assess the community pharmacists' readiness to implement the deprescribing of inappropriate medications for older adults in Qatar. In addition, the secondary objective of the study was to Identify the significant predictors of community pharmacists' readiness.

## Methods

### Study design, setting and sampling

A theory-informed cross-sectional survey of community pharmacists' readiness to implement the deprescribing of inappropriate medications for older adults was conducted between 01 November, 2023 and 20 January 2024 in Qatar. This is a small Middle Eastern Gulf country with an estimated population of 2.73 million, and most of whom are expatriates [29].

The purposive sample of community pharmacists who participated in the cross-sectional survey was drawn from a sampling frame that was based on the list obtained from the Ministry of Public Health in Qatar. This list included all licensed community pharmacists working in Qatar. The inclusion criteria used for sampling of the study participants were all licensed community pharmacists in practice for at least a year in Qatar, and who were able to communicate orally and in writing in English language. The choice of English language was due to the fact that the community pharmacy sector in Qatar is dominated by foreigners who are able to speak and write in English. In addition, sampling of study participants was conducted with the proportional distribution of community pharmacists working in chain pharmacies in Qatar, and this was based on the information obtained from the list provided by the Ministry of Public Health. The calculation of the minimum sample of 252 community pharmacists required for the study was done with Raosoft online calculator and an extra 10% was added to adjust for non-response. The parameters used for the calculation of the required sample of 252 include following: target population of community pharmacists (562), margin of error (5%), confidence level (95%), and response distribution (50%). The finite population correction was not considered during the sample size calculation.

### Questionnaire development and structure

The development of the questionnaire was informed by the readiness components of the situational theory of leadership, and an initial draft of 50 questions was prepared after a thorough review of relevant literature in the research area

[15–20,24]. This was followed by an iterative process involving an in-depth discussion of the appropriateness, relevance and validity of the items by the research team. This resulted in the final 40-item questionnaire comprising 3 sections including A (demographics), B (knowledge of the purpose/goals and process of deprescribing, and the screening toolkits and algorithms used for deprescribing), and C (confidence to implement deprescribing in practice). A panel of three senior researchers with relevant experience in the clinical pharmacy and practice research area assessed the content validity of the questionnaire. The relevance of each of items included in the questionnaire was determined by the panel, and disagreements were resolved through consensus. In addition, reliability analysis of the final questionnaire was determined with the internal consistency method using Cronbach alpha, and these were 0.71 and 0.73 for sections B and C respectively. Lastly, the questionnaire was pre-tested on a sample of 10 community pharmacists that were selected through simple random sampling to assess clarity, completeness and understanding. The final questionnaire was modified as necessary, based on the pre-test result that was not included in the final results.

The study participants were asked to rank their responses to the items in Section B on a 2-point Likert-type scale (true = 1, false = 0, and don't know = 0). The responses to the items included in Section C were done with a 4-point Likert-type scale (strongly agree = 4, agree = 3, disagree = 2, strongly disagree = 1). The maximum obtainable score for confidence was 6 while that of knowledge was 10. Hence, the maximum score for community pharmacists' readiness was 16.

### Data collection and analysis

Data collection was done with the self-administered validated and pre-tested 40-item questionnaire. The distribution and collection of the completed questionnaire were done by three research assistants, and the procedure used for the distribution and collection was standardized with a prior training to minimize inter-individual variation. The questionnaires were delivered to the 252 sampled community pharmacists at their workplaces, and detailed information including the purpose, scope and anticipated benefits of the study were provided. The signed informed consent form was obtained before the start of data collection. Participants' right to decline participation were upheld and data collection was anonymous. Completed self-administered questionnaires were collected as soon possible but not exceeding 5 days after distribution. Reminders were sent to respondents who were unable to return completed questionnaires within one week through telephone calls or emails.

The IBM SPSS (Statistics for Windows, Version 29.0. Armonk, NY: IBM Corp.) statistical software was used for data analysis. Shapiro-Wilk test was used to test the normality of community pharmacists' readiness score (confidence + knowledge). Descriptive statistics such as median (IQR) was used for continuous data with non-normal distribution while frequency (%) was used for categorical data. The predictors of community pharmacists' readiness to implement deprescribing were identified with the use of binomial logistic regression analysis that commenced with an initial check to ensure that the key assumptions for a valid binary logistic regression analysis were met. These include the use of a dichotomous scale of measurement for the dependent variable, independent observations, and the absence of multiple collinearities among independent variables. The alpha level for significance was set at $p \leq 0.05$.

### Ethics

Ethics approval was obtained from QUIRB (Qatar University Institutional Review Board) (QU-IRB 1906-E/23, August 30, 2023).

### Results

Out of the 252 community pharmacists sampled and who consented, two hundred completed the questionnaires (response rate, 79.4%). The demographic and practice characteristics of the respondents are as shown in Table 1. The readiness scores showed non-normal distribution (Shapiro Wilk test, 0.93, $p < 0.001$). The majority of the community pharmacists were females (n = 109, 54.5%), within the age range of 20–40 years (n = 176, 88.0%), had BSc / BPharm as

**Table 1. Community pharmacists' demographic and practice characteristics (N = 200).**

| Item | n (%) |
|---|---|
| **Gender** | |
| Male | 91 (45.5) |
| Female | 109 (54.5) |
| **Age group (years)** | |
| 20-30 | 102 (51.0) |
| 31-40 | 74 (37.0) |
| 41-50 | 24 (12.0) |
| **Nationality** | |
| Indian | 81 (40.5) |
| Egyptian | 45 (22.5) |
| Jordanian | 44 (22.0) |
| Sudanese | 12 (6.0) |
| Pakistani | 7 (3.5) |
| Filipino | 4 (2.0) |
| Syrian | 3 (1.5) |
| Saudi | 2 (1.0) |
| Palestinian | 1 (0.5) |
| Ghana | 1 (0.5) |
| **Highest educational Level** | |
| BSc/BPharm | 141 (70.5) |
| MSc / MPharm | 33 (16.5) |
| PharmD | 26 (13.0) |
| **Employment status** | |
| Full time | 194 (97.0) |
| Part time | 6 (3.0) |
| **Years of experience as a community pharmacist** | |
| 1-5 years | 103 (51.5) |
| 6-10 years | 57 (28.5) |
| >10 years | 40 (20.0) |
| **Previous hospital pharmacy experience** | |
| Yes | 110 (55.0) |
| No | 90 (45.0) |
| **Exposure to deprescribing in undergraduate training** | |
| Yes | 98 (49.0) |
| No | 102 (51.0) |
| **Attended CPD related to deprescribing in past 5 years** | |
| None | 108 (54.0) |
| 1-2 | 56 (28.0) |
| 3-4 | 18 (9.0) |
| >4 | 18 (9.0) |
| **Willingness to complete CPD on deprescribing (Median (IQR))** | 4 (3, 5) |

the highest educational qualification (n = 141, 70.5%), were full-time employee (n = 194, 97.0%). Furthermore, the majority of the respondents had 1–5 years of community pharmacy experience (n = 103, 51.5%) and previous hospital pharmacy experience (n = 110, 55%), but only 49% (98) reported exposure to deprescribing in their undergraduate training. In addition, the majority of respondents reported not attending any CPD related to deprescribing in the past 5 years (n = 108, 54%). However, the median (IQR) score for the respondents' willingness to complete a CPD program on deprescribing of medicines for older adults was 4 (2) (maximum = 5) (Table 1).

Community pharmacists' knowledge of deprescribing and its purpose, and the toolkits and algorithms used in practice are shown in Table 2. Overall, the community pharmacists' median (IQR) knowledge score was 7 (2) (minimum = 1, maximum = 10). The majority of the respondents correctly identified the fact that deprescribing is not focused only on medication stoppage (n = 115, 57.5%), or initiated only for patients experiencing adverse side effects (n = 144, 72%) or during the use of only prescription-only medicines (n = 155, 77.5%) but it involves other key interventions such as dosage reduction, medication stoppage or switching as necessary (n = 167, 83.5%). However, a sizeable proportion of community pharmacists were unable to correctly define deprescribing (n = 85, 42.5%), or correctly identify if it is initiated only when adverse side effects were encountered by patients (n = 56, 28%).

Furthermore, the majority of the community pharmacists were knowledgeable about the toolkits used for deprescribing varieties of medicines class such as proton pump inhibitors and benzodiazepines (n = 106, 53%), and antipsychotics and cholinesterase inhibitors (n = 108, 54%). In addition, the majority of respondents reported the awareness of the algorithms used for deprescribing inappropriate medications in older adults (n = 106, 53%), and are knowledgeable about to how to access these relevant deprescribing resources (n = 127, 63.5%). In addition, the majority of community pharmacists opined that the provision of a deprescribing service will reduce the risk of exposure to adverse side effects (n = 177, 88.5%), and maximize medication adherence for older adults (n = 184, 92%).

Community pharmacists' confidence to implement a deprescribing service in Qatar is shown in Table 3. The median (IQR) confidence score is 6 (2) (minimum = 1, maximum = 6). The majority of respondents were confident to identify inappropriate medications that will require deprescribing in older adults (n = 173, 86.5%), and also considered the provision of a deprescribing service as an important role of community pharmacists (n = 185, 92.5%). Furthermore, the majority of the respondents feel confident to discuss deprescribing opportunities in practice with physicians (n = 175, 87.5%) and patients

**Table 2. Community pharmacists' knowledge of deprescribing, and its purpose, toolkits and algorithm (N = 200).**

| Item | True n (%) | False n (%) | Don't Know n (%) |
|---|---|---|---|
| Deprescribing is a term that solely describes stoppage of a medication (**False**) | 70 (35.0) | **115 (57.5)** | 15 (7.5) |
| Deprescribing involves either stoppage of a medication, dosage reduction or switching from one drug class to another (**True**) | **167 (83.5)** | 28 (14.0) | 5 (2.5) |
| Deprescribing of a medication is initiated only when a patient is experiencing side-effects (**False**) | 44 (22.0) | **144 (72.0)** | 12 (6.0) |
| Deprescribing toolkits exist for the Proton pump inhibitors and benzodiazepines receptor agonists. (**True**) | **106 (53.0)** | 33 (16.5) | 61 (30.5) |
| Deprescribing toolkits exist for the cholinesterase inhibitors and antipsychotics (**True**) | **108 (54.0)** | 35 (17.5) | 57 (28.5) |
| Awareness of the algorithms used for the deprescribing of inappropriate medications in older adults (**True**) | **106 (53.0)** | 28 (14.0) | 66 (33.0) |
| Deprescribing is only applicable to 'Prescription Only Medications (POM)' and does not apply to OTC medications (**False**) | 34 (17.0) | **155 (77.5)** | 11 (5.5) |
| Deprescribing will help to reduce the risk of patients 'exposure to adverse events due to drug therapy (**True**) | **177 (88.5)** | 13 (6.5) | 10 (5.0) |
| Deprescribing will help to improve patient's adherence to medication regimen | **184 (92.0)** | 5 (2.5) | 11 (5.5) |
| I know how to access the relevant resources needed to support deprescribing | **127 (63.5)** | 33 (16.5) | 39 (19.5) |

**Table 3. Community Pharmacists' attitude and confidence to implement deprescribing for older adults in Qatar (N = 200).**

| Item | Strongly agree n (%) | Agree n (%) | Disagree n (%) | Strongly Disagree n (%) |
|---|---|---|---|---|
| Deprescribing inappropriate medications is an important role of community pharmacists. | 125 (62.5) | 60 (30.0) | 13 (6.5) | 2 (1.0) |
| I am confident to identify medications that may require deprescribing. | 65 (32.5) | 108 (54.0) | 27 (13.5) | 0 (0%) |
| I am confident to discuss opportunities for deprescribing with physicians. | 59 (29.5) | 116 (58.0) | 22 (11.0) | 3 (91.5) |
| I am confident to discuss opportunities for deprescribing with patients. | 82 (41.0) | 98 (49.0) | 17 (8.5) | 3 (1.5) |
| My pharmacy training prepared me to discuss deprescribing opportunities with the physicians and patients. | 49 (24.5) | 90 (45.0) | 52 (26.0) | 9 (4.5) |
| Learning about how to implement deprescribing in my practice setting is a top priority for my CPD needs. | 67 (33.5) | 110 (55) | 16 (8.0) | 7 (3.5) |

(n = 180, 90%). In addition, the majority of respondents feel that continuous learning on the implementation of an effective deprescribing service is a top CPD priority (n = 177, 88.5%).

The median (IQR) score for the readiness of community pharmacists to implement deprescribing of inappropriate medications for older adults was 13 (4) (minimum = 4, maximum = 16). Overall, 54.4% (109) of community pharmacists reported high confidence score (≥ median), while 60.5% (121) reported high knowledge score (≥ median). However, the proportion of community pharmacists adjudged based on the theoretical framework of the situational theory of leadership as having high confidence + high knowledge to implement deprescribing of inappropriate medications for older adults was 41%, followed by those with low confidence + low knowledge (26%), low confidence + high knowledge (19.5%) and high confidence + low knowledge (13.5%).

Lastly, the binary logistic regression analysis showed that the significant predictors of community pharmacists' readiness to implement deprescribing for older adults were gender (female) (OR = 2.75, 95%CI: 1.22–4.34, p = 0.0048), and exposure to curricular contents related to deprescribing of inappropriate medicines during undergraduate training (OR = 3.85, 95%CI: 2.61–5.13, p = 0.001) (Table 4).

## Discussion

Overall, the majority of the community pharmacists felt knowledgeable and confident to implement deprescribing for older adults in Qatar. This is because the median readiness score approximates 81% of the maximum obtainable score. Indeed, the majority of respondents correctly describe deprescribing, identified the appropriate contexts for its use and the required resources; and the clinical benefits inherent in implementing deprescribing especially for older adults. These findings bode well for involving community pharmacists in the delivery of deprescribing service in Qatar, and the results further add to the published literature in the research area that showed that community pharmacists possess the required competence to effectively provide deprescribing service that improves patient outcomes especially for older adults [22–25].

The current study showed that the most significant predictors of community pharmacists' readiness to implement deprescribing were female gender and exposure to the concept of deprescribing during undergraduate training. These predictors are being reported for the first time, and seems unsurprising as the female gender has been identified by previous studies as a significant predictor of community pharmacists' self-perceived competency to provide a clinically-related service such as managing minor ailments [30–31]. It is not readily clear what factors may be underlining the reported better self-perceived readiness to implement deprescribing among female community pharmacists, and this is probably a good lead for further research. However, female community pharmacists have been shown to be better at gaining patients' trust through the effective use of interpersonal communication skills that enable them to build good rapport [32]. Perhaps, this may have contributed to enhancing their self-efficacy and perceived readiness to implement a clinical service such as

**Table 4. Binary logistic regression of the predictors of community pharmacists' readiness to implement deprescribing for older adults (N = 200).**

| Item | Categories (n) | Readiness | | B | SE | Wald | Exp(B) | 95% CI for Exp(B) | | P-value |
|---|---|---|---|---|---|---|---|---|---|---|
| | | Median (<13) n (%) | Median (≥13) n (%) | | | | | Lower | Upper | |
| Gender | Male (91) | 59 (64.8) | 32 (35.2) | | | | 1(reference) | | | |
| | Female (109) | 39 (35.8) | 70 (64.2) | 0.793 | 0.235 | 7.861 | 2.75 | 1.22 | 4.34 | 0.0048* |
| | | | | | | | | | | |
| Age groups (years) | > 40 (176) | 71 (40.3) | 105 (59.7) | | | | 1(reference) | | | |
| | ≤ 40 (24) | 11(45.8) | 13 (54.2) | 0.430 | 0.312 | 1.07 | 1.69 | 0.35 | 2.63 | 0.213 |
| | | | | | | | | | | |
| Highest pharmacy degree | BSc/BPharm (141) | 66 (46.8) | 116 (53.2) | | | | 1(reference) | | | |
| | Non-BSc/BPharm (59) | 33 (55.9) | 26 (44.1) | 0.74 | 0.65 | 2.57 | 0.71 | 0.39 | 1.46 | 0.237 |
| | | | | | | | | | | |
| Employment status | Full time (194) | 85 (43.8) | 109 (56.2) | | | | 1(reference) | | | |
| | Part time (6) | 3 (50.0) | 3 (50.0) | 0.53 | 0.46 | 1.62 | 0.75 | 0.47 | 1.87 | 0.583 |
| | | | | | | | | | | |
| Years of experience | ≤ 5 (103) | 55 (53.4) | 48 (46.6) | | | | 1(reference) | | | |
| | >5 (97) | 48 (49.5) | 49 (50.5) | 0.56 | 0.60 | 0.13 | 0.82 | 0.45 | 1.91 | 0.340 |
| | | | | | | | | | | |
| Previous hospital pharmacy experience | Yes (110) | 62 (56.4) | 48 (43.6) | | | | 1(reference) | | | |
| | No (90) | 47 (52.2) | 43 (47.8) | 0.52 | 0.41 | 1.57 | 0.67 | 0.41 | 1.84 | 0.468 |
| | | | | | | | | | | |
| Exposure to deprescribing in UG | No (102) | 56 (54.9) | 46 (45.1) | | | | 1(reference) | | | |
| | Yes (98) | 32 (32.7) | 66 (67.3) | 0.87 | 0.274 | 6.78 | 3.85 | 2.61 | 5.13 | 0.001* |
| | | | | | | | | | | |
| Attended CPD to deprescribing (years) | ≤ 1 (164) | 83 (50.6) | 81 (59.8) | | | | 1(reference) | | | |
| | >5 (36) | 15 (41.7) | 21 (58.3) | 0.35 | 0.03 | 2.25 | 0.58 | 0.29 | 1.71 | 0.143 |

NB: 13 (median score) is the cut-off point for readiness (maximum is 16); B = Coefficient; SE = Standard Error; Exp(B) = Exponentiation of coefficient; CI = Confidence Interval; * p < 0.05 (statistically significant), UG = Undergraduate.

deprescribing. This is plausible as higher perceived self-efficacy is a well-documented factor that enhances self-perceived competency or readiness to successfully complete assigned tasks [33–34].

The literature is scanty on the impact of exposure to deprescribing during undergraduate pharmacy training on actual practice. However, Clark et al reported an increased self-efficacy to implement deprescribing in practice when exposed to the flipped classroom pedagogical approach to teach deprescribing and safe medication use in a third-year pharmacotherapy course [35]. Therefore, giving a strong consideration to involving female community pharmacists and those who had been exposed to deprescribing in their undergraduate training at the initial stage of the development and piloting of a deprescribing service scheme might prove useful especially in Qatar, and similar settings, where such a service is currently not within the scope of community pharmacy practice. However, these findings are in contrast to that of Heinrich et al, who reported that demographic factors did not influence community pharmacists' knowledge of deprescribing in Ireland [25]. Perhaps, the observed contrast might be due to differences in settings, and the scope and level of community pharmacy practice. This is because the community pharmacy sector in Qatar is dominated mainly by foreigners with diverse training and practice experience obtained mainly from developing countries, and where service orientation is mainly product-oriented [36].

Notwithstanding the moderately high median readiness score observed among community pharmacists, a more nuanced analysis showed critical knowledge gaps that will require educational intervention to improve readiness. For instance, only about two-fifth of the respondents could be classified into the high knowledge and high confidence group based on the theoretical framework used for the current study. In addition, about half of the respondents admitted their lack of knowledge of deprescribing resources such as toolkits and algorithms; and how to access and use them. These findings are unsurprising as at least half of the respondents reported that they were neither exposed to the deprescribing content in their undergraduate training nor attended any CPD related to deprescribing in the past five years. These findings are consistent with that of El-Dahiyat et al, who reported that only 25.3% of community pharmacists in UAE had good knowledge of deprescribing. This finding is probably a reflection of the diverse nature of pharmacy workforce in the Middle Eastern Gulf countries as the community pharmacy sector is dominated mainly by foreigners from developing countries with exposure to different pharmacy curriculums that may or may not have adequate contents related to deprescribing, and different practice experience that is yet to meet the expanded scope obtainable in developed settings where community pharmacy-led deprescribing has been implemented [30–31]. Furthermore, the current scope of practice for community pharmacists in Qatar does not include the provision of a clinically-oriented service such as deprescribing. Hence, targeted capacity-building educational interventions are warranted to fill these competency gaps. Thankfully, this is a plausible proposition as the majority of respondents reported a high willingness to complete a CPD program focused on deprescribing; and about 90% of community pharmacists considered continuous learning about how to implement deprescribing in their practice as a top CPD priority. These findings clearly bode well for the development and implementation of a CPD intervention focused on improving the readiness of the community pharmacists to implement deprescribing in Qatar. In addition, these findings are probably a good index of community pharmacists' aspiration to expand their roles within the health system in Qatar. This seems plausible as their assumption of the task of deprescribing will provide the opportunity to enhance their contribution to positive patient outcomes especially at the primary care level [24,37,38].

## Strengths and limitations

To best of our knowledge, the current study is the first to foreground the assessment of readiness of community pharmacists to implement deprescribing with a holistic theoretical framework that defined readiness as consisting of two key components including knowledge and confidence. However, the interpretation of the study findings may be limited by few factors including the use of a non-probability sampling method as the study participants were drawn mainly from the chain pharmacies. Hence, the reported results may not fully capture the view of all community pharmacists in Qatar. However, chain pharmacies constitute about 75% of the community pharmacies in Qatar [39]. Therefore, the reported findings are probably a good reflection of the current reality. In addition, the use of the purposive sampling method may have confounded the observed readiness score as 54.5% of the study participants were females. However, published studies have shown female community pharmacists to be better at gaining patient trust through the effective use of interpersonal communication skills that enable them to build good rapport.[32] Perhaps, this may have contributed to enhancing their self-efficacy and perceived readiness to implement a clinical service such as deprescribing [32–33]. Furthermore, the use of the quantitative survey method and the fact that deprescribing as a clinical service guided by a structured formal framework is currently not within the scope of practice for community pharmacists may have increased the risk of social desirability / self-report bias as the community pharmacists who responded may have done so in a favorable manner leading to inflated self-reported knowledge and confidence scores. In addition, the self-administered nature of the questionnaires which were left behind may have allowed some respondents to consult information resources especially for the knowledge-focused items, and this may increase the risk of overestimation of knowledge scores. However, this appeared not be the case as a sizeable proportion of the respondents reported low scores for a sizeable proportion of the items used for the assessment of knowledge.

## Conclusions

Overall, the majority of the community pharmacists felt knowledgeable and confident to implement deprescribing of inappropriate medicines for older adults in a developing setting such as Qatar. These findings bode well for involving community pharmacists in the delivery of a deprescribing service for older adults. However, a number of critical readiness gaps especially regarding how to access and use deprescribing toolkits and algorithms were identified, and these gaps require educational interventions to improve community pharmacists' readiness. The most significant predictors of community pharmacists' readiness score were female gender and exposure to deprescribing in undergraduate training.

## Author contributions

**Conceptualization:** Kazeem Babatunde Yusuff.

**Data curation:** Kazeem Babatunde Yusuff, Marwa Elshazly, Sondus Jawad, Ayesha Ahmed, Hager ElGeed.

**Formal analysis:** Kazeem Babatunde Yusuff, Marwa Elshazly, Sondus Jawad, Ayesha Ahmed, Hager ElGeed.

**Funding acquisition:** Kazeem Babatunde Yusuff.

**Methodology:** Kazeem Babatunde Yusuff, Marwa Elshazly, Sondus Jawad, Ayesha Ahmed, Hager ElGeed.

**Project administration:** Kazeem Babatunde Yusuff.

**Resources:** Kazeem Babatunde Yusuff.

**Software:** Kazeem Babatunde Yusuff.

**Supervision:** Kazeem Babatunde Yusuff, Hager ElGeed.

**Validation:** Kazeem Babatunde Yusuff, Sondus Jawad, Ayesha Ahmed.

**Visualization:** Kazeem Babatunde Yusuff, Marwa Elshazly, Sondus Jawad, Ayesha Ahmed, Hager ElGeed.

**Writing – original draft:** Kazeem Babatunde Yusuff, Marwa Elshazly, Sondus Jawad, Ayesha Ahmed.

**Writing – review & editing:** Kazeem Babatunde Yusuff, Hager ElGeed.

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
