## [Decision Letter · Decision Letter 0]

18 Jun 2025

Dear Dr. Yusuff,

Thank you for submitting your manuscript to PLOS ONE. After careful consideration, we feel that it has merit but does not fully meet PLOS ONE’s publication criteria as it currently stands. Therefore, we invite you to submit a revised version of the manuscript that addresses the points raised during the review process.

We look forward to receiving your revised manuscript.

Kind regards,

Ilhem Berrou, PhD

Academic Editor

PLOS ONE

Journal Requirements:

[This work was supported by an Undergraduate Research Experience Program (UREP) award [UREP29-092-3-029] from the Qatar National Research Fund (a member of The Qatar Foundation). The contents are solely the responsibility of the authors.].

Additional Editor Comments:

Thank you for submitting this interesting and well written manuscript.

In addition to the reviewers comments, please also address the following:

Introduction and literature review

Please shorten this section, succinctly highlighting: the problem of polypharmacy, its prevalence in Qatar, the Middle East comparing to global patterns, why it needs to be addressed including any data relevant to Qatar e.g. prevalence of medicines related problems and how deprescribing can address them.

Clearly defining deprescribing and how it can be done by community pharmacists. Many deprescribing interventions happen already, informally, over the counter in some countries. In others, deprescribing happens more formally following formal protocols and under specific job titles.

What are the barriers to community pharmacists deprescribing in Qatar and other countries.

Please justify the choice of theory.

Discussion

Please shorten this section be more succinct. Please discuss key findings (predictors of readiness to de-prescribe and levels of knowledge) in relation to what was reported in other studies, taking care to succinctly highlight the relevant to Qatari context.

It will be useful to highlight how this learning may feed into a future deprescribing service in community pharmacy in Qatar.

It is noted that the pharmacy workforce is diverse, presented per nationality in the results section. What does this mean in relation to the findings e.g. practices pharmacists bring from their home country.

As a limitation it is important to address the limitations of answering questions about a service that is yet to exist.

Reviewers' comments:

Reviewer's Responses to Questions

**Comments to the Author**

1. Is the manuscript technically sound, and do the data support the conclusions?

Reviewer #1: Yes

2. Has the statistical analysis been performed appropriately and rigorously?

Reviewer #1: Yes

3. Have the authors made all data underlying the findings in their manuscript fully available?

Reviewer #1: Yes

4. Is the manuscript presented in an intelligible fashion and written in standard English?

Reviewer #1: Yes

Reviewer #1: Thank you for your submission of this research article on the knowledge, confidence and readiness of community pharmacists to implement deprescribing in Qatar which I really enjoyed reading and would be happy to see published pending minor revision

You clearly highlighted the rationale for undertaking this research by using a theoretical framework and undertaking it in a developing country which does not yet have a community pharmacy deprescribing programme. Related to this (and the discussion on page 14 where it is suggested that results may differ by settings), consider briefly describing the scope of community pharmacists in Qatar for the international audience since this differs internationally i.e. Can they prescribe independently? Would they be able to stop/change medicines independently or would this be through a doctor?

The sample size calculation, questionnaire development/testing and statistical approaches (descriptive statistics of IQR/frequency and logistic regression) are all clearly described/justified.

The results of the three questionnaires included support the conclusion drawn that the majority of community pharmacists felt knowledgeable and confident. Related to these results, when Table 2 is described it is said that "However, a sizeable proportion of community pharmacists incorrectly described deprescribing as solely involving medication stoppage (42.5%, 85/200), or initiated only when adverse side effects are encountered by patients (28%; 56/200)" - however these frequencies combine those who answered incorrectly and those who answered don't know - hence rather than saying they incorrectly described deprescribing it would be more correct to say that they were "unable to correctly define".

The logistic regression included supports the conclusion drawn that female sex and exposure to deprescribing at undergraduate level were predictors of readiness to deprescribe. While the consideration of female sex in the literature is discussed, it is not for the other predictor - is there any relevant literature on the impact of deprescribing teaching in undergraduate pharmacy programmes that this could be linked to?

The manuscript is clearly presented and readable but would benefit from editing as there are minor typographical and grammatical errors throughout - some examples include missing words like "a", changing from singular to plural within the same sentence (or vice versa) and some inconsistencies in Tables with missing brackets and sometimes within the same column using a decimal point for a whole number and other times not.

**Do you want your identity to be public for this peer review?** For information about this choice, including consent withdrawal, please see our Privacy Policy

Reviewer #1: No

---

## [Author Response · Author response to Decision Letter 1]

9 Jul 2025

9 July 2025

The Editor-In-Chief

PLOS ONE

Dear Sir,

Re: Manuscript #PONE- D-25-19691 – “Predictors of community pharmacists’ readiness to implement deprescribing of inappropriate medications for older adults in Qatar”

Our sincere thanks for the opportunity to revise the manuscript #PONE-D-25-19691, titled “Predictors of community pharmacists’ readiness to implement deprescribing of inappropriate medications for older adults in Qatar” which is under your consideration for publication in the PLOS ONE. We thank the reviewers for the insightful comments and useful suggestions and we have revised the manuscript accordingly. Please find stated below our point-by-point response to the reviewers’ comments. We have also revised the manuscript in accordance with the editor’s comments.

EDITOR’S COMMENTS

Journal Requirements:

1. When submitting your revision, we need you to address these additional requirements. Please ensure that your manuscript meets PLOS ONE's style requirements, including those for file naming. The PLOS ONE style templates can be found at https://journals.plos.org/plosone/s/file?id=wjVg/PLOSOne_formatting_sample_main_body.pdf and

Response: The corrections have been done in accordance with specifications stated in the PLOS ONE style template

KBY

Undergraduate Research Experience Program (UREP) award [UREP29-092-3-029] from the Qatar National Research Fund (a member of The Qatar Foundation). The contents are solely the responsibility of the authors. Please state what role the funders took in the study. If the funders had no role, please state: "The funders had no role in study design, data collection and analysis, decision to publish, or preparation of the manuscript." If this statement is not correct you must amend it as needed. Please include this amended Role of Funder statement in your cover letter; we will change the online submission form on your behalf.

Response: The financial disclosure statement has been amended as recommended, and also included in the revised cover letter.

Response: The references have been reviewed and revised as necessary.

Additional Editor Comments:

Introduction and literature review:

1. Please shorten this section, succinctly highlighting: the problem of polypharmacy, its prevalence in Qatar, the Middle East comparing to global patterns, why it needs to be addressed including any data relevant to Qatar e.g. prevalence of medicines related problems and how deprescribing can address them. Clearly defining deprescribing and how it can be done by community pharmacists. Many deprescribing interventions happen already, informally, over the counter in some countries. In others, deprescribing happens more formally following formal protocols and under specific job titles.

Response: The corrections have been done as recommended.

2. What are the barriers to community pharmacists deprescribing in Qatar and other countries.

Response: The manuscript has been revised to highlight the fact that the barriers have been assessed in a separate study that was previously published PLOS ONE.

3. Please justify the choice of theory.

Response: The manuscript has been revised to enhance the clarity of details of the justification for the choice of the theory.

Discussion

4. Please shorten this section be more succinct. Please discuss key findings (predictors of readiness to deprescribe and levels of knowledge) in relation to what was reported in other studies, taking care to succinctly highlight the relevant to Qatari context. It will be useful to highlight how this learning may feed into a future deprescribing service in community pharmacy in Qatar.

Response: The discussion has been revised accordingly.

5. It is noted that the pharmacy workforce is diverse, presented per nationality in the results section. What does this mean in relation to the findings e.g. practices pharmacists bring from their home country.

Response: The corrections have been done as recommended.

6. As a limitation it is important to address the limitations of answering questions about a service that is yet to exist.

Response: This has been addressed in the revised manuscript as recommended.

REVIEWER’S COMMENTS

Reviewer #1: Thank you for your submission of this research article on the knowledge, confidence and readiness of community pharmacists to implement deprescribing in Qatar which I really enjoyed reading and would be happy to see published pending minor revision Response: We are truly grateful for the valuable comments offered by the reviewer and the time spent to provide these useful feedback. We value the suggested corrections proposed by the reviewer and we are convinced it can only improve the scholarly value of the manuscript.

• Comment-1: You clearly highlighted the rationale for undertaking this research by using a theoretical framework and undertaking it in a developing country which does not yet have a community pharmacy deprescribing programme. Related to this (and the discussion on page 14 where it is suggested that results may differ by settings), consider briefly describing the scope of community pharmacists in Qatar for the international audience since this differs internationally i.e. Can they prescribe independently? Would they be able to stop/change medicines independently or would this be through a doctor?

• Response-1: We thank the reviewer for this excellent suggestion and we totally concur with the premise of the suggested correction and it has been done. [Pg 15, 1st paragraph, line 4-7, 2nd Paragraph, 12-18].

• Comment-2: The sample size calculation, questionnaire development/testing and statistical approaches (descriptive statistics of IQR/frequency and logistic regression) are all clearly described/justified.

• Response-2: We are truly grateful for the valuable comments offered by the reviewer and the time spent to provide these useful and kind feedback.

• Comment-3: The results of the three questionnaires included support the conclusion drawn that the majority of community pharmacists felt knowledgeable and confident. Related to these results, when Table 2 is described it is said that "However, a sizeable proportion of community pharmacists incorrectly described deprescribing as solely involving medication stoppage (42.5%, 85/200), or initiated only when adverse side effects are encountered by patients (28%; 56/200)" - however these frequencies combine those who answered incorrectly and those who answered don't know - hence rather than saying they incorrectly described deprescribing it would be more correct to say that they were "unable to correctly define"

• Response-3: This is an excellent suggestion and we sincerely thank the reviewer. The revision has been done as recommended [Pg 10, 1st paragraph, line 8-11].

• Comment-4: The logistic regression included supports the conclusion drawn that female sex and exposure to deprescribing at undergraduate level were predictors of readiness to deprescribe. While the consideration of female sex in the literature is discussed, it is not for the other predictor - is there any relevant literature on the impact of deprescribing teaching in undergraduate pharmacy programmes that this could be linked to?

• Response-4: Heartfelt thanks to the reviewer for the valuable comment. The revision has been done as recommended [Pg 14, paragraph 2, line 15-20].

• Comment-5: The manuscript is clearly presented and readable but would benefit from editing as there are minor typographical and grammatical errors throughout - some examples include missing words like "a", changing from singular to plural within the same sentence (or vice versa) and some inconsistencies in Tables with missing brackets and sometimes within the same column using a decimal point for a whole number and other times not.

• Response-5: Heartfelt thanks to the reviewer for the suggestions. The corrections have been done as recommended.

---

## [Decision Letter · Decision Letter 1]

9 Oct 2025

Dear Dr. Yusuff,

Thank you for submitting your manuscript to PLOS ONE. After careful consideration, we feel that it has merit but does not fully meet PLOS ONE’s publication criteria as it currently stands. Therefore, we invite you to submit a revised version of the manuscript that addresses the points raised during the review process.

In the manuscript, you reported that the questionnaire was informed by the situational theory of leadership; however, this was not clearly reflected in the questionnaire development section. You also indicated that the sampling technique used was *purposive sampling* . From your description, it appears that participants may have been recruited primarily based on ease of access and availability. This approach is generally referred to as *convenience sampling* rather than purposive sampling.<o:p></o:p>

Please carefully revisit your Methods section to ensure that the sampling technique is accurately described. If you intended purposive sampling, please clarify the specific inclusion criteria and rationale that guided participant selection.<o:p></o:p>

In the Results section, you have reported frequencies in the format *(percentage, n/total sample size)* . For consistency and clarity, please revise the reporting style to the standard format *(n = XX, percentage%)* instead of *(n/total sample size)* . For example, please use *(n = 109, 54.5%)* instead of *(109/200, 54.5%)* .<o:p></o:p>

We look forward to receiving your revised manuscript.

Kind regards,

Mohammad Nusair, Ph.D

Academic Editor

PLOS ONE

Journal Requirements:

Additional Editor Comments:

In the manuscript, you reported that the questionnaire was informed by the situational theory of leadership; however, this was not clearly reflected in the questionnaire development section. You also indicated that the sampling technique used was purposive sampling. From your description, it appears that participants may have been recruited primarily based on ease of access and availability. This approach is generally referred to as convenience sampling rather than purposive sampling.

Please carefully revisit your Methods section to ensure that the sampling technique is accurately described. If you intended purposive sampling, please clarify the specific inclusion criteria and rationale that guided participant selection.

In the Results section, you have reported frequencies in the format (percentage, n/total sample size). For consistency and clarity, please revise the reporting style to the standard format (n = XX, percentage%) instead of (n/total sample size). For example, please use (n = 109, 54.5%) instead of (109/200, 54.5%).

Reviewers' comments:

Reviewer's Responses to Questions

**Comments to the Author**

Reviewer #1: (No Response)

Reviewer #2: All comments have been addressed

2. Is the manuscript technically sound, and do the data support the conclusions?

Reviewer #1: Yes

Reviewer #2: Yes

3. Has the statistical analysis been performed appropriately and rigorously?

Reviewer #1: Yes

Reviewer #2: Yes

4. Have the authors made all data underlying the findings in their manuscript fully available?

Reviewer #1: Yes

Reviewer #2: Yes

5. Is the manuscript presented in an intelligible fashion and written in standard English?

Reviewer #1: Yes

Reviewer #2: Yes

Reviewer #1: Thank you for submitting your revised manuscript which you have clearly put a lot of work into which has strengthened your paper further

Just a couple of comments based on your amendments

Your response -1: We thank the reviewer for this excellent suggestion and we totally concur with the premise of the suggested correction and it has been done. [Pg 15, 1st paragraph, line 4-7, 2nd Paragraph, 12-18]

Unfortunately didn't make it any clearer for me (or perhaps other international readers) what the the scope of community pharmacists in Qatar is i.e. Can they prescribe independently? Would they be able to stop/change medicines independently or would this only be through a doctor? Please clarify

There are minor typographical and grammatical errors throughout still - again examples include missing words like "a" or "the", changing from singular to plural within the same sentence (or vice versa) and some inconsistencies in Tables etc but the manuscript is still easily readable/understandable. Specific ones I would suggest to change based on your submitted amendments are:

Abstract Line 4 should be "A" theory as at the start of the sentence not "a" theory

Abstract Line 7 - remove "a" before Qatar

Introduction Paragraph 3 - not having two sequential sentences starting with "indeed"

Table 1 CPD - None not to one decimal place like others

Table 2 - Extra bracket in "True" header

References - Not all in same format - one example would be 35 has no year listed for a journal article and not all formatting guidelines have been followed since reference citations are requested to be in brackets in the text as described https://journals.plos.org/plosone/s/file?id=wjVg/PLOSOne_formatting_sample_main_body.pdf

Reviewer #2: The article is really informative and the topic 'deprescribing' is interesting.

The manuscript covered the basic research structure.

Honestly, I cant find anything to be corrected.

Well done for the research team.

**Do you want your identity to be public for this peer review?** For information about this choice, including consent withdrawal, please see our Privacy Policy

Reviewer #1: No

Reviewer #2: **Yes:** Nasser M Alorfi

---

## [Author Response · Author response to Decision Letter 2]

21 Nov 2025

19 Nov 2025

The Editor-In-Chief

PLOS ONE

Dear Sir,

Re: Manuscript #PONE- D-25-19691_R1 – “Predictors of community pharmacists’ readiness to implement deprescribing of inappropriate medications for older adults in Qatar”

Our sincere thanks for the opportunity to revise the manuscript #PONE-D-25-19691_R1, titled “Predictors of community pharmacists’ readiness to implement deprescribing of inappropriate medications for older adults in Qatar” which is under your consideration for publication in the PLOS ONE. We thank the reviewers for the insightful comments and useful suggestions and we have revised the manuscript accordingly. Please find stated below our point-by-point response to the reviewers’ and editor’s comments. We have also revised the manuscript in accordance with the editor’s comments.

EDITOR’S COMMENTS

Journal Requirements:

Response: There was no such recommendation from the reviewers

Response: The reference list has been reviewed and the suggested corrections have been done. There was no incidence of retraction.

3. In the manuscript, you reported that the questionnaire was informed by the situational theory of leadership; however, this was not clearly reflected in the questionnaire development section.

Response: The manuscript has been revised accordingly [Pg 6, 3rd para, line 1-3].

4. You also indicated that the sampling technique used was purposive sampling. From your description, it appears that participants may have been recruited primarily based on ease of access and availability. This approach is generally referred to as convenience sampling rather than purposive sampling. Please carefully revisit your Methods section to ensure that the sampling technique is accurately described. If you intended purposive sampling, please clarify the specific inclusion criteria and rationale that guided participant selection.

Response: The purposive sampling method was used and the method section has been revised to enhance the clarity of the procedure used for sampling as recommended [Pg 6, 2nd para, line 1-6].

5. In the Results section, you have reported frequencies in the format (percentage, n/total sample size). For consistency and clarity, please revise the reporting style to the standard format (n = XX, percentage%) instead of (n/total sample size). For example, please use (n = 109, 54.5%) instead of (109/200, 54.5%).

Response: The result section has been revised as recommended [Pg 9-12].

REVIEWER’S COMMENTS

Reviewer #1: Thank you for submitting your revised manuscript which you have clearly put a lot of work into which has strengthened your paper further.

Response: We are truly grateful for the valuable comments offered by the reviewer and the time spent to provide these useful feedback. We value the suggested additional corrections and we are convinced it can only improve the scholarly value of the manuscript.

• Comment-1: Your response -1: We thank the reviewer for this excellent suggestion and we totally concur with the premise of the suggested correction and it has been done. [Pg 15, 1st paragraph, line 4-7, 2nd Paragraph, 12-18]. Unfortunately, didn't make it any clearer for me (or perhaps other international readers) what the scope of community pharmacists in Qatar is i.e. Can they prescribe independently? Would they be able to stop/change medicines independently or would this only be through a doctor? Please clarify.

• Response-1: We thank the reviewer for this suggestion. The manuscript has been revised to emphasize the fact that the scope of practice for community pharmacists does not currently include the provision of deprescribing service in Qatar. [Pg 15, 2nd para, line 18-20].

• Comment-2: There are minor typographical and grammatical errors throughout still - again examples include missing words like "a" or "the", changing from singular to plural within the same sentence (or vice versa) and some inconsistencies in Tables etc but the manuscript is still easily readable/understandable. Specific ones I would suggest to change based on your submitted amendments are:

• Abstract Line 4 should be "A" theory as at the start of the sentence not "a" theory

• Abstract Line 7 - remove "a" before Qatar

• Introduction Paragraph 3 - not having two sequential sentences starting with "indeed"

• Table 1 CPD - None not to one decimal place like others

• Table 2 - Extra bracket in "True" header

• Response-2: We are truly grateful for these valuable comments offered by the reviewer and the time spent to provide these useful and kind feedback. All the recommended corrections have been done.

• Comment-3: References - Not all in same format - one example would be 35 has no year listed for a journal article and not all formatting guidelines have been followed since reference citations are requested to be in brackets in the text as described https://journals.plos.org/plosone/s/file?id=wjVg/PLOSOne_formatting_sample_main_body.pdf

• Response-3: This is an excellent suggestion and we sincerely thank the reviewer. The in-text citations and the reference list have been revised as recommended.

Reviewer #2: The article is really informative and the topic 'deprescribing' is interesting.

The manuscript covered the basic research structure. Honestly, I cant find anything to be corrected.

Well done for the research team.

Response: Heartfelt thanks for the valuable comments offered by the reviewer. We feel greatly encouraged, and for that we are truly grateful.

---

## [Decision Letter · Decision Letter 2]

9 Dec 2025

Dear Dr. Yusuff,

**The reviewers are pleased with the revised manuscript and provided positive recommendations. However, one of the reviewers has asked for minor revisions that I would like you to address before making a final decision on your submission.**

We look forward to receiving your revised manuscript.

Kind regards,

Mohammad Nusair, Ph.D

Academic Editor

PLOS One

**Journal Requirements:**

Reviewers' comments:

Reviewer's Responses to Questions

**Comments to the Author**

Reviewer #1: All comments have been addressed

Reviewer #2: All comments have been addressed

2. Is the manuscript technically sound, and do the data support the conclusions?

Reviewer #1: (No Response)

Reviewer #2: Yes

3. Has the statistical analysis been performed appropriately and rigorously?

Reviewer #1: (No Response)

Reviewer #2: Yes

4. Have the authors made all data underlying the findings in their manuscript fully available?

Reviewer #1: (No Response)

Reviewer #2: (No Response)

5. Is the manuscript presented in an intelligible fashion and written in standard English?

Reviewer #1: (No Response)

Reviewer #2: (No Response)

**Reviewer #1:** (No Response)

**Reviewer #2:** English Proof reading is required.

I strongly recommend a structural abstract ( Background, method etc .. ). Also, I recommend adding one sentence ( about Self-Report Bias into the limitation section ) that this may lead to inflated confidence scores would make it clearer.

Overall, the manuscript is strong, well-revised, and suitable for publication

**Do you want your identity to be public for this peer review?** For information about this choice, including consent withdrawal, please see our Privacy Policy

Reviewer #1: No

Reviewer #2: **Yes:** Nasser M Alorfi

---

## [Author Response · Author response to Decision Letter 3]

15 Jan 2026

15-01-2026

The Editor-In-Chief

PLOS ONE

Dear Sir,

Re: Manuscript #PONE- D-25-19691R2 – “Predictors of community pharmacists’ readiness to implement deprescribing of inappropriate medications for older adults in Qatar”

Our sincere thanks for the opportunity to revise the manuscript #PONE-D-25-19691R1, titled “Predictors of community pharmacists’ readiness to implement deprescribing of inappropriate medications for older adults in Qatar” which is under your consideration for publication in the PLOS ONE. We thank the reviewers for the useful suggestions and we have revised the manuscript accordingly. Please find stated below our point-by-point response to the reviewers’ and editor’s comments. We have also revised the manuscript in accordance with the editor’s comments.

REVIEWER’S COMMENTS

Reviewer #2:

Comment-1: I strongly recommend a structural abstract ( Background, method etc .. ).

Response-1: We are thankful for the suggestion offered by the reviewer. The abstract was prepared based on the template / requirements provided by PLOS One and we thought we should strictly adhere to these.

Comment-2: I recommend adding one sentence (about Self-Report Bias into the limitation section) that this may lead to inflated confidence scores would make it clearer.

Response-2: We are truly grateful for this valuable suggestion offered by the reviewer. The issue of self-report bias identified by the reviewer was addressed under self-desirability bias in the “strength and limitation” section. However, the manuscript was revised to enhance its clarity in compliance with the reviewer’s suggestion [page 16 line 16-19, page 17, line 1-7].

---

## [Decision Letter · Decision Letter 3]

28 Jan 2026

Predictors of community pharmacists’ readiness to implement deprescribing of inappropriate medications for older adults in Qatar

PONE-D-25-19691R3

Dear Dr. Yusuff,

We’re pleased to inform you that your manuscript has been judged scientifically suitable for publication and will be formally accepted for publication once it meets all outstanding technical requirements.

Kind regards,

Mohammad Nusair, Ph.D

Academic Editor

PLOS One

Additional Editor Comments (optional):

Reviewers' comments:

Reviewer's Responses to Questions

**Comments to the Author**

Reviewer #1: All comments have been addressed

Reviewer #2: All comments have been addressed

2. Is the manuscript technically sound, and do the data support the conclusions?

Reviewer #1: (No Response)

Reviewer #2: Yes

3. Has the statistical analysis been performed appropriately and rigorously?

Reviewer #1: (No Response)

Reviewer #2: Yes

4. Have the authors made all data underlying the findings in their manuscript fully available?

Reviewer #1: (No Response)

Reviewer #2: Yes

5. Is the manuscript presented in an intelligible fashion and written in standard English?

Reviewer #1: (No Response)

Reviewer #2: Yes

Reviewer #1: (No Response)

Reviewer #2: My comments about ''Predictors of community pharmacists’ readiness to implement deprescribing of inappropriate medications for older adults in Qatar'' were sorted. The manuscript is ready for acceptance.

**Do you want your identity to be public for this peer review?** For information about this choice, including consent withdrawal, please see our Privacy Policy

Reviewer #1: No

Reviewer #2: **Yes:** Nasser M Alorfi

---

## [Editor Report · Acceptance letter]

PONE-D-25-19691R3

PLOS One

Dear Dr. Yusuff,

I'm pleased to inform you that your manuscript has been deemed suitable for publication in PLOS One. Congratulations! Your manuscript is now being handed over to our production team.

Kind regards,

on behalf of

Dr. Mohammad Nusair

Academic Editor

PLOS One